Gram staining reveals diverse bacterial associations in coral cell-associated microbial aggregates in the Pacific Ocean

Singhakarn Chutimon chutimon@hawaii.edu 1 2
Toonen Robert J. 1
Work Thierry M. 2
1 Hawai‘ i Institute of Marine Biology, University of Hawai‘ i at Mānoa , Kāne‘ohe , Hawai‘ i , United States of America
2 Honolulu Field Station, US Geological Survey, National Wildlife Health Center , Honolulu , Hawai‘ i , United States of America
Pogoreutz Claudia
Electronic publication date: 2025 Aug 19
Publication date: 2025
Volume: 13
Electronic Location ID: e19867
Received 2025 Jan 10; Accepted 2025 Jul 17
Copyright: ©2025 Singhakarn et al.
Copyright year: 2025
Copyright holder: Singhakarn et al.
License: This is an open access article distributed under the terms of the Creative Commons Attribution License, which permits unrestricted use, distribution, reproduction and adaptation in any medium and for any purpose provided that it is properly attributed. For attribution, the original author(s), title, publication source (PeerJ) and either DOI or URL of the article must be cited.
License URL: https://creativecommons.org/licenses/by/4.0/

Keywords: Tissue distribution, Porites, Pocillopora, Acropora, Hawaii, Coral microbiome

Funding: US Geological Survey This work was funded by US Geological Survey. The funders had no role in study design, data collection and analysis, decision to publish, or preparation of the manuscript.

==============================
Cell-associated microbial aggregates (CAMAs) (also referred to as coral-associated microbial aggregates) have been observed in 24 coral species from the Pacific Ocean, and studies indicate most contain gram-negative bacilli from the genus Endozoicomonas. Here, we used histology with Gram staining to evaluate the morphology and distribution of CAMAs in six species of scleractinian corals from Hawaii and Palmyra. Within CAMAs, we observed the coexistence of bacteria with differing morphologies and Gram-staining properties both within and among coral species. Pocillopora and Acropora had mostly gram-negative bacilli, whereas gram-negative cocci dominated in Porites. Acropora had the highest abundance of gram-positive CAMAs. The anatomical distribution of CAMAs varied by coral species. CAMAs dominated in the tentacles of Pocillopora meandrina, Pocillopora grandis, and Porites evermanni, were mostly in the coenenchyme of Acropora cytherea, and were found equally between tentacles and coenenchyme in Porites compressa and Porites lobata. Tissue-layer distribution also varied, with CAMAs mainly in the epidermis of Pocillopora but in the gastrodermis of Porites and Acropora. The diversity of bacteria in CAMAs and their anatomic distribution in Pacific corals may be more complex than previously understood. This indicates other bacterial species, in addition to Endozoicomonas, are colonizing CAMAs in corals from the Pacific Ocean.

Introduction

Microbes within the surface mucus layer, coral tissue, and skeleton play a vital role in coral holobiont function (Siboni et al., 2008; Van Oppen & Blackall, 2019; Tandon et al., 2022; Mohamed et al., 2023). Although substantial knowledge exists on microbial metagenomics of mucus (Lee et al., 2015; Glasl, Herndl & Frade, 2016; Hadaidi et al., 2017), relatively less is known about the identity, location, or function of the internal microbiome of corals (Wada et al., 2019; Maire et al., 2023; Maire et al., 2024). As such, interest is growing surrounding bacteria-coral interactions involving cell-associated microbial aggregates (CAMAs), also referred to as “coral-associated microbial aggregates” (Wada et al., 2019; Wada et al., 2022). First observed in Acropora palmata (Peters, Oprandy & Yevich, 1983) and Porites astreoides (Peters, 1984), CAMAs are now recognized in 24 coral species across the Pacific Ocean, and are particularly prevalent in the genera Acropora, Porites, and Pocillopora (Work & Aeby, 2014). While common among healthy individuals, CAMAs are decreased or absent in Porites experiencing tissue loss (Sudek et al., 2012). The high prevalence of CAMAs in some coral tissues not associated with host cell pathology indicates they may play a role in coral health similar to endosymbionts (Work & Aeby, 2014).

Based on molecular and morphological studies of Stylophora pistillata, the genus Endozoicomonas is the dominant bacterium in CAMAs for that species of coral globally (Neave et al., 2017). Using molecular assays and fluorescence in situ hybridization (FISH), Endozoicomonas has been confirmed in tissues of S. pistillata from the Red Sea (Bayer et al., 2014), the western Pacific (Wada et al., 2022), and Micronesia (Neave et al., 2016). However, it is likely that other microbes are involved in CAMAs formation and function of CAMAs. For example, in Acropora hyacinthus, five distinct bacterial morphologies were seen in different CAMAs using fluorescent in situ hybridization (FISH) and 16S rRNA probes: rod-shaped, atypical coccus, longer rod morphology, filamentous-like bacteria, and rod-shaped morphology with spore-like structure (Wada et al., 2019). In contrast, bacterial morphology using transmission electron microscopy was similar within and between CAMAs in Pocillopora acuta (Maire et al., 2023). There is also evidence of coexistence of multiple bacterial taxa within a single CAMA. For example, Simkania and Endozoicomonas were identified from CAMA samples using laser microdissection and 16S ribosomal RNA gene metabarcoding (Maire et al., 2023). Application of FISH further demonstrated co-localization within a given aggregate, thereby underscoring CAMA complexity. Three Endozoicomonas metagenomes recovered from CAMAs indicate the potential for bacteria synthesizing antioxidants, antimicrobial compounds, and several B vitamins, which may be essential for coral and Symbiodiniaceae health (Maire et al., 2023). However, sorting out how these functions relate to coral health still needs further experimental clarification.

Knowledge of the distribution and diversity of bacteria within CAMAs is important to understanding their role in coral health. While molecular methods like metagenomics and in situ hybridization are powerful tools, they have limitations. Metagenomics indicates the presence or absence of DNA but gives no information on whether bacteria are intact, their morphology, or anatomical location. In situ hybridization localizes organisms to tissues, but requires a priori knowledge of the organisms to be identified so that proper probes can be designed. In the absence of such knowledge, when presented with unknown bacteria in tissues, diagnosticians use histological techniques coupled with Gram staining (Gram, 1884; Wilson et al., 2015) to guide laboratory investigations. Gram stains categorize bacteria as gram-positive (blue to purple) or gram-negative (red to pink) based on cell wall properties (Smith & Hussey, 2005). Knowing the gram status of bacteria in tissues allows for efficient downstream application of appropriate confirmatory diagnostic steps. For example, the presence of gram-negative bacteria in tissues might lead to the use of culture media selective for growth of that bacterial type for isolation and further characterization (Jung & Hoilat, 2024).

Here, we used Gram stains to evaluate CAMAs in three genera of scleractinian corals (Acropora, Pocillopora, and Porites) that were commonly found to host these microbial aggregates in previous surveys (Work & Aeby, 2014). These coral genera represent three of the most important reef-building families globally (Acroporidae, Pocilloporidae, and Poritidae) comprising different evolutionary lineages and life history strategies (Siqueira, Kiessling & Bellwood, 2022; Jury et al., 2024). These three families together comprise roughly 95% of coral cover throughout the Hawaiian Archipelago (Franklin, Jokiel & Donahue, 2013). In Hawaii, Acropora has a restricted geographic distribution in the Northwestern Hawaiian Islands and is rare from the Main Hawaiian Islands (Walsh et al., 2014; Concepcion et al., 2016). In contrast, Pocillopora is a common early colonizer on reefs, whereas Porites is a slower-growing massive coral most common on more established reefs throughout the Hawaiian Archipelago. Comparing the composition and distribution of CAMAs across these genera provides a reasonable sampling of scleractinian diversity to assess location and composition of CAMAs among reef-building coral taxa.

Materials & Methods

Coral identification was based on growth form, corallites, and patterns of bumps or rounded projections following descriptions from Wells et al. (1988). Acropora cytherea colonies are tabulate with fine, upward-projecting branchlets and radial corallites with short open calices and a terminal corallite. Pocillopora grandis colonies have stout upright flattened branches with tubercles interspersed with distinct corallites. Pocillopora meandrina colonies are similar to Pocillopora grandis but colonies comprise short branches radiating from a central area. Porites compressa branches are distinct to fused cylindrical forms with closely apposed small corallites. Porites lobata colonies are usually hemispherical or lobed and may be more than four meters wide with a smooth surface and closely apposed corallites and are yellow to pale brown. Porites evermanni colonies are similar but dark brown.

Samples were collected between 2001 and 2021 from the Main and Northwestern Hawaiian Islands, including Island of Hawai‘ i, O‘ ahu, Kaua‘ i, Kānemiloha‘ i (French Frigate Shoals), Nalukākala (Maro Reef), and Palmyra Atoll (Fig. 1, Table 1) under Hawaii Department of Aquatic Resources (Permit: SAP2025-28). During collections, colonies were photographed, and gross lesions were categorized as apparently normal, algae overgrowth, bleaching, discoloration, tissue loss, or growth anomaly (Work & Aeby, 2006, Fig. 2A). Coral fragments were fixed with Z-Fix (Anatech, Battle Creek, MI, USA) diluted 1:5 with seawater and decalcified using Cal-Ex II (Thermo Fisher Scientific, Waltham, MA, USA), trimmed, and processed following standard histological methods as described (Work & Aeby, 2010). Sections were recut onto glass slides and stained with the modified Brown and Hopps method (Schwartz et al., 1989) (referred to hereafter as Gram stain). Gram stains were performed by Wisconsin Veterinary Diagnostic Laboratory, a laboratory certified by the USDA National Animal Health Laboratory Network (US Food and Drug Administration, 2013). In gram-positive bacterial cells, the thick peptidoglycan layer in the cell wall prevents the elution of the crystal violet–iodine complex during decolorization, allowing these cells to retain the stain turning them purple-blue (Erkmen, 2021). In contrast, gram-negative cells have a thinner peptidoglycan layer, which allows the crystal violet–iodine complex to diffuse out during decolorization, leaving the cells visible only after being counterstained with safranin that turns them red.

Figure 1 A map showing the location of coral collection sites in the Pacific Island region including Island of Hawai‘ i, O‘ ahu, Kaua‘ i, Kānemiloha‘ i (French Frigate Shoals-FFS), Nalukākala (Maro Reef) and Palmyra Atoll.

Table 1 Number of fragments sampled for histology partitioned by location and species.

Depths at the sampled sites are 3–10 m.

Species	Kānemiloha‘ i (French Frigate Shoals)	Island of Hawai‘ i	O‘ ahu	Kaua‘ i	Nalukākala (Maro Reef)	Palmyra Atoll	Total	
Pocillopora grandis	1			12			13	
Pocillopora meandrina		8	1	12			21	
Porites compressa	3	4	13				20	
Porites evermanni	5		3				8	
Porites lobata	1	3	15		1		20	
Acropora cytherea						5	5	
Total	10	15	32	24	1	5	87	

Figure 2 Number of cell-associated microbial aggregates (CAMAs) among coral species.

(A) Percentage of gross lesion types within coral species. (B) Number of CAMAs per cm2 in individual fragments by corals species. The colors of points indicate gross lesion in each fragment. The box extent shows the middle 50% of number of CAMAs/cm2 in individual coral specie. The upper and lower whiskers show upper 25% and lower 25% of number of CAMAs/cm2 excluding outlier in individual coral species. Middle line and red number indicates median CAMA/cm2 of each coral species. Poc., Pocillopora, Por., Porites; and A., Acropora.

Some bacteria exhibit gram-variable staining, meaning gram-positive bacteria may stain as gram-negative. This can occur when certain gram-positive bacteria, such as Bacillus, Butyrivibrio, and Clostridium have a thinner cell wall during exponential growth phase (Beveridge, 2001) or a damaged cell wall (Popescu & Doyle, 1996) thereby not allowing retention of crystal violet-iodine complex. The most common mistake is misidentifying gram-positive bacteria as gram-negative due to excessive decolorization, which causes the cells to appear gram-negative (Popescu & Doyle, 1996). To minimize such errors, we included a known gram-positive and gram-negative controls in our staining procedure to verify our results. Our gram-negative control was kidney tissue from a White-tailed tropicbird (Phaethon lepturus) with lesions of salmonellosis and presence of small gram-negative bacilli from which pure cultures of Salmonella typhimurium were isolated. Our gram-positive control was liver tissue from a Laysan duck (Anas laysanensis) with lesions of sepsis and presence of large gram-positive bacilli from which pure cultures of Erysipelothrix rhusiopathiae were isolated. Tissues were fixed in 10% neutral buffered formalin, trimmed, and embedded in paraffin, sectioned at five µm, and stained with Gram stain by Wisconsin Veterinary Diagnostic Laboratory.

Two sets of coral histological slides were examined by a single observer. The first set was a total of 87 fragments from 76 colonies comprising six coral species including Acropora cytherea (n = 4), Pocillopora grandis (n = 8), Pocillopora meandrina (n = 16), Porites compressa (n = 20), Porites lobata (n = 20), and Porites evermanni (n = 8) (File S1). This set was used to investigate whether the bacterial morphologies and gram status within CAMAs, their anatomical and tissue layer location, and their density (CAMAs per cm2 of tissue) are consistent across different Hawaiian reef-building coral species, lesion types, and sites. The second set of samples comprised paired normal and lesion fragments from which we enumerated CAMAs per tissue area (cm2) from paired apparently healthy and lesion fragments of 35 diseased coral colonies. This set was used to assess whether the density of CAMAs (per cm2 of tissue) differs between apparently healthy and lesioned fragments within the same colony. We used a subset of previously examined histology slides (n = 52) and added 23 additional slides that were paired with the original slides from the same individual colonies (File S2). Some samples were excluded from the first set of histological slides due to the absence of a complete tissue pair (either normal or lesion).

Slides were scanned using HP Color LaserJet pro MFP M479fdw (HP Inc., Palo Alto, CA, USA). The scanned images were used only to calculate the surface area (cm2) of tissue from individual fragments using QuPath version 0.4.3 (Bankhead et al., 2017). The assessment of color or Gram stain was done during using BX43 compound microscope (Olympus corporation, Waltham, MA, USA), and by images of CAMA from Gram-stained slides that were taken with an INFINITY3 digital microscope camera (Lumenera Corporation, Ottawa, Ontario, Canada). Brightness, contrast, and hue were set at zero. The images were taken at 100X magnification. Distribution of CAMA was categorized by anatomical location (skeleton, tentacles, mesentery, coenenchyme (Peters, 2016)). Within tissues, CAMA was localized to specific layers including epidermis, mesoglea, gastrodermis, and calicodermis. Bacterial shape within CAMA was classified as bacilli and cocci. CAMAs staining red to pink on Gram stain were classified as gram-negative while those staining purple-to-blue were classified as gram-positive. CAMAs whose bacteria were densely packed and whose shape could not be clearly categorized were categorized as undetermined and excluded from further analyses (n = 150, 11.17% of total observed CAMAs). The number of CAMAs from each individual fragment was normalized by surface tissue area (cm2) of histology slide to account for differences in the amounts of tissue examined on each slide.

All analyses were done with RStudio version 2,023.06.1+524 (Posit Team, 2023). Points in box plot/violin plot were jittered, where individual data points are added to the box plot with each point being slightly shifted randomly to avoid overlap, to enhance clarity of presentation (Cleveland, 1985). The number of CAMAs per cm2 did not fit parametric assumptions of normal distribution and equal variance, so they were compared between coral species, gross lesion, and location using the Kruskal–Wallis test. Percentages of gram status and bacterial shape were compared between species, anatomical location and tissue layer using Pearson’s chi-squared test and pairwise proportion tests (post-hoc). The number of CAMAs per tissue area (cm2) was compared between apparently normal and lesion fragments from the same colony using the Wilcoxon signed-rank test. All statistics were run using the ggstatplot package version 0.13.0 (Patil, 2021), and rstatix package version 0.7.2 (Kassambara, 2023). Shannon’s index (Shannon, 1948) was used to compare the diversity of bacterial morphologies observed per coral species with the R package vegan version 2.6-10 (Oksanen et al., 2022).

Results

Of 87 fragments across six species of coral, we identified a total of 1,310 CAMAs. The highest median number of CAMAs/cm2 per fragment was 7.62 in Porites compressa (inter quartile range (IQR) = 0.102) followed by Pocillopora meandrina (median = 6.88, IQR = 0.121), Pocillopora grandis (median = 6.84, IQR = 0.192), Porites evermanni (median = 5.96, IQR = 0.033), A. cytherea (median = 4.17, IQR = 0.036) and Porites lobata (median = 2.38, IQR = 0.032). There was no significant difference in the median number of CAMAs/cm2 between coral species (Kruskal–Wallis test, p = 0.072) (Fig. 2B). Median number of CAMAs/cm2 by lesion type ranged from 1.01 for algae overgrowth to 10.45 for bleaching. Number of CAMAs/cm2 was not significantly different among five gross lesion types (Kruskal–Wallis test, p = 0.097) (Fig. 3). Median (IQR) of CAMAs/cm2 per fragment between location were 13.2 (0.106) for the Island of Hawai‘ i, 6.59 (0.167) for Kaua‘ i, 4.98 (0) for Nalukākala (Maro Reef), 4.83 (0.051) for Kānemiloha‘ i (French Frigate Shoals), 4.17 (0.036) for Palmyra, and 3.44 (0.058) for O‘ ahu. Number of CAMAs/cm2 differed significantly by location (Kruskal–Wallis test, p-value = 0.0095) with corals from Island of Hawai‘ i having significantly greater numbers of CAMAs compared to those from O‘ ahu (p = 0.0057) (Fig. 4). There was no significant difference (Wilcoxon signed-rank test, p = 0.17) in the number of CAMAs/cm2 between paired normal and lesion fragments from individual diseased colonies (n = 35) (Fig. 5).

Figure 3 Boxplot for number of cell-associated microbial aggregates (CAMAs) per area (cm2) by gross lesions.

The box extent shows the middle 50% of number of CAMAs/cm2 by gross lesion. The upper and lower whiskers show upper 25% and lower 25% of number of CAMAs/cm2 excluding outlier in individual gross lesion. Middle line and red number indicates median number of CAMA/cm2 across gross lesion. Points indicate outlier data. Algae overgrowth has the lowest median number 1.01 of CAMAs/cm2. Bleaching has the highest median number 10.45 of CAMAs/cm2. Example of gross lesion (margin indicated by arrowhead) (A) algae overgrowth in Pocillopora meandrina (B) bleaching in Porites lobata (C) discoloration in Porites evermanni (D) growth anomaly in Porites compressa; note corals exhibiting excessive growth of skeleton in relation to adjacent polyps on the same colony (E & F) a tissue loss in Pocillopora meandrina and Acropora cytherea, respectively.

Figure 4 Boxplot/violin plots showing the number of cell-associated microbial aggregates (CAMAs) per area (cm2) across different locations.

The box extent shows the middle 50% of number of CAMAs/cm2 by locations. The upper and lower whiskers show upper 25% and lower 25% of Number of CAMAs/cm2 excluding outlier in individual locations. Middle line and red dot indicates median number of CAMAs/cm2 across locations. The violin width shows frequency of the value, the wider sections indicating higher frequency of the value of CAMAs/cm2. Corals from Island of Hawai‘ i had significantly more CAMAs/cm2 than those from O‘ ahu (Kruskal–Wallis test, p = 0.0095; Dunn’s post-hoc test with Holm correction, p = 0.00567).

Figure 5 Boxplot/violin plots showing cell-associated microbial aggregates (CAMAs) per area (cm2) in paired apparently normal and lesion fragments from diseased coral colonies (n = 35).

The box extent shows the middle 50% of number of CAMA/cm2 by status of fragments. The upper and lower whiskers show upper 25% and lower 25% of number of CAMAs/cm2, excluding outliner in individual status. Middle line indicates median number of CAMA/cm2 across status. The violin width shows frequency of the value, the wider sections indicating higher frequency of the value of CAMAs/cm2. There was no significant difference in CAMAs per area between normal and lesion fragments (Wilcoxon signed-rank test, p = 0.17).

Anatomical and tissue layer distributions of CAMAs vary among coral species

The distribution of CAMAs within anatomical compartments and tissue layers varied among coral species. In Pocillopora meandrina, Pocillopora grandis, and Porites evermanii, most CAMAs were located in the tentacles, accounting for 96.5%, 95.5%, and 74.6% of anatomical compartments, respectively. A. cytherea had CAMAs mainly in the coenenchyme. In Porites compressa and Porites lobata, CAMAs were equally distributed between the tentacles and coenenchyme. Three CAMAs were observed in the skeleton, an inanimate portion of the coral in an area where skeleton had presumably been removed during decalcification (Fig. 6A). For tissue layers, CAMAs in Pocillopora were mainly in the epidermis (92.7% and 93.9% for Pocillopora grandis and Pocillopora meandrina, respectively; in Porites CAMAs were mainly in the gastrodermis. CAMAs in A. cytherea were mainly in the gastrodermis and mesoglea with relatively fewer in the calicodermis (Fig. 6B).

Figure 6 Distribution of cell-associated microbial aggregates (CAMAs) by (A) anatomical location (B) tissue layers.

(A) Contrast preponderance of bacteria in coenenchyme of Acropora cytherea, Porites compressa, and Porites lobata compared to tentacles for Pocillopora spp. and Porites evermanni. (B) Note dominance of epidermal bacterial colonization in Pocillopora spp. in contrast to gastrodermis for Porites spp. and A. cytherea. (n, total CAMA count). Poc., Pocillopora, Por., Porites, and A., Acropora.

Bacterial morphology and gram staining

Of 1,310 CAMAs, we were able to confidently determine bacterial shape and Gram-staining status for 1,193. CAMAs had four bacterial morphologies: gram-negative bacilli (Fig. 7A), gram-negative cocci (Figs. 7B–7C), gram-positive bacilli (Fig. 7D), and gram-positive cocci (Fig. 7E). Avian tissues infected with Salmonella typhimurium showed small gram-negative bacilli (Fig. 7F) while avian tissues infected with Erysipelothrix rhusiopathiae showed larger gram-positive bacilli (Fig. 7G). In corals, CAMAs were dominated by gram-negative bacteria except for A. cytherea where 36% of CAMAs were gram-positive (Pearson’s chi-squared, p = 0.1). Bacteria within CAMAs in A. cytherea, Pocillopora meandrina and Pocillopora grandis were bacillus-shaped whereas those in CAMAs from Porites compressa, Porites evermanni, and Porites lobata were mostly coccoid with a minority being bacillus-shaped bacteria (13%, 8% and 7%, respectively) (Fig. 8A). We observed coexistence of bacteria differing in both bacterial shape and Gram-staining characteristics between individual CAMAs in 34% of fragments where morphology of CAMAs could be reliably identified. The diversity of bacterial morphologies was highest in A. cytherea with a Shannon index of 0.768, followed by Porites compressa (0.471), Porites meandrina (0.402), Porites lobata (0.397), Porites evermanni (0.267), and Pocillopora grandis (0.241) (Fig. 8B).

Figure 7 Gram-stain images of cell-associated microbial aggregates (CAMAs) from various Pacific Ocean corals, and reference Gram-stain images of infectious bacterial agents confirmed by culture and molecular techniques.

(A) Gram-negative bacilli clustered (black arrow) and diffusely distributed (white arrowhead) in the epidermis and gastrodermis of the tentacle of Pocillopora grandis from Kaua‘ i. (B) Gram-negative cocci (arrow) in the coenosteum of Porites lobata from O‘ ahu. (C) Gram-negative cocci (arrow) in the gastrodermis of Porites compressa from O‘ ahu. (D) Gram-positive bacilli (arrow) in the basal body wall gastrodermis of Porites compressa from the Island of Hawai‘ i. (E) Gram-positive cocci CAMAs (arrow) in the gastrodermis of Porites lobata from the Island of Hawai‘ i. Abbreviations: g, gastrodermis; e, epidermis; sk, skeleton; z, symbiodiniaceae; black arrowhead, calicodermis; arrow, CAMAs. (F) cluster of Salmonella typhimurium, gram-negative bacilli (black arrow) in the kidney of White-tailed tropicbird (Phaeton lepturus). (G) Erysipelothrix rhusiopathiae, gram-positive bacilli focally distributed (black arrow) in the kidney of a Laysan duck (Anas laysanensis). Scale bar = 10 µm (A & D) and 30 µm for all other plates.

Figure 8 Morphology and gram status of cell-associated microbial aggregates (CAMAs) by (A) coral species and (B) individual coral fragment.

(A) Note that gram-negative bacilli dominate for Porites (Por.), gram-negative coccoid dominate for Pocillopora (Poc.), and Acropora (A.) are distinguished by relatively high abundance of gram-positive bacteria (n, total CAMA count). (B) Note low diversity of bacteria morphologies in Porites evermmani.

Discussion

This study aimed to characterize the abundance, morphology, and Gram-staining status of CAMAs across different Hawaiian reef-building coral genera and lesion types, in both diseased and visually normal tissues using histology with Gram staining. In contrast to Sudek et al. (2012), who found a 74% reduction in the number of CAMAs in Porites colonies with the condition bleaching and tissue loss versus healthy colonies, we found no significant difference in the number of CAMAs per cm2 between visually normal and lesioned fragments from the same individual diseased colonies. This discrepancy could be due to differences in sample type, as our study focused solely on fragments from diseased colonies, a recognized limitation of our study. Also, we did not examine lesions of bleaching with tissue loss, so perhaps there are particular responses of CAMAs with certain lesion types. For instance, we saw that corals with algal overgrowth exhibited fewer CAMAs compared to corals with other types of gross lesions. Algal overgrowth on corals associated with phase shifts driven by nutrient overload is a well-recognized issue in coral reef ecosystems (Bell & Elmetri, 1995; Done, 1992; McCook, 1999). Molecular studies have shown that nutrient pollution and increased algal cover can shift bacterial communities, with an increase in opportunistic Proteobacteria and a decrease in Actinobacteria within the surface mucus layer of corals (Haas et al., 2016; Zaneveld et al., 2016). Although the relationship between nutrient levels, algal overgrowth and CAMA abundance remains unclear, it may warrant further investigation. Temporal studies of CAMAs abundance could help determine whether declines in CAMAs precede or follow algal overgrowth, and sampling only normal colonies would be useful to assess the status of CAMA in clinically normal Pocillopora, Porites and Acropora.

Most CAMAs in Pocillopora were found in the epidermis of the tentacles, while in Porites evermanni, Porites lobata, and Porites compressa, they were primarily located in the gastrodermis of the tentacles and coenenchyme. This distribution may reflect the anatomy of these corals, as more complex perforate corals like Acropora and Porites (Okubo, 2016) may provide a greater variety of habitats and cell types available for bacterial colonization. Similarly, in Acropora, CAMAs were predominantly found in the coenenchyme, which is abundant between widely spaced corallites (Fenner, 2005). These contrast with non-perforate Pocillopora where CAMAs were almost exclusively in epidermis of the tentacles. In other invertebrates, the location of microbial aggregations is often linked to specific physiological functions of the host. For example, Aliivibrio fischeri in squid aggregate in the cilia of its light organ behind the gut-ink sac, thereby facilitating bioluminescence during nighttime activity (Nyholm & McFall-Ngai, 2021). Similarly, Wolbachia bacteria in mosquitoes preferentially localize in specific regions of the oocyte during oogenesis, enabling them to infect the female germline and induce parthenogenesis, turning unfertilized eggs into diploid females (Correa & Ballard, 2016). In corals, the specific localization of CAMAs could similarly influence physiological functions. For example, CAMAs in the gastrodermis may play a role in nutrient absorption and digestion, while those in the epidermis might affect interactions with the external environment or contribute to host defense. Investigating how the anatomical location of CAMAs relates to coral function could provide valuable insights into their role in coral health.

Although DNA sequencing is commonly used to identify microbial communities, Gram staining remains a valuable tool for initial surveys and has a long history of use in diagnostic pathology. As such, our study complements existing studies on CAMA by providing important information of localizing organism to anatomical location. For example, most CAMAs comprised gram-negative bacilli, morphologically consistent with the genus Endozoicomonas, which would accord with other studies that show this bacterium to be widely abundant across various marine invertebrates and fish (Neave et al., 2016; Pogoreutz & Ziegler, 2024). Endozoicomonas has been identified from CAMAs in Stylophora pistillata, Pocillopora verrucosa, and Pocillopora acuta (Bayer et al., 2014; Neave et al., 2017; Maire et al., 2023). However, the presence of gram-positive bacilli and cocci and gram-negative cocci indicates additional species of bacteria exist in at least five species of Pacific corals. Gram-positive bacteria in coral tissues have been extensively documented using both culture-dependent (Sweet et al., 2021) and culture-independent (Ainsworth et al., 2015) methods. For example, A. cytherea hosts a small proportion of gram-positive cocci bacteria (<1%) identified as Candidatus actinomarina (Qin et al., 2022), a morphology similar to what was observed here.

A. cytherea exhibited the greatest diversity of CAMAs, with multiple morphologies and Gram staining. Other authors have also noted varying morphologies of bacteria in CAMAs from Acropora. For example, Wada et al. (2019) found different morphologies of CAMA bacteria in A. hyacinthus from Australia. This variability of bacteria within a CAMA could indicate organisms that share metabolic byproducts, a phenomenon known as bacterial cross-feeding reviewed by Smith et al. (2019). Bacterial diversity between and within CAMA could also potentially enhance host resilience akin to algal endosymbionts where different genera can vary spatially and temporally within individual coral colonies (Rouzé et al., 2019; Rocha de Souza et al., 2023) and this diversity enhances resilience during environmental stress (Cunning, Silverstein & Baker, 2018). Analogously, microbial diversity within CAMAs may also be a strategy that helps corals adapt to environmental perturbations. Understanding the specific bacterial species within these CAMAs and exactly how they interact with the host would be crucial to understand their role in coral immunity and health. Future research could focus on methods to grow these CAMA in culture (Sweet et al., 2021), laser capture microdissection to identify targeted CAMAs by molecular means (Maire, Blackall & Van Oppen, 2021; Maire et al., 2023; Maire et al., 2024), eliminating or introducing CAMAs into corals experimentally and monitoring host fitness (Palincsar et al., 1989; Schuett et al., 2007), or assessing viability of bacteria using methods like propidium monoazide (Nocker et al., 2007).

Extracellular gram-negative bacilli were observed in all tissue layers of the tentacles of Pocillopora grandis, as well as in the skeletons, in areas where skeleton was removed during decalcification, of A. cytherea and Porites lobata. This could indicate the rupture of the double-layer membrane surrounding bacterial aggregates (Palincsar et al., 1989). These CAMA bacteria might infect neighboring cells through intercellular spread, similar to obligate intracellular bacteria like Rickettsia spp. (Van Schaik et al., 2013), or they may persist outside host cells as facultative bacteria (Maire et al., 2023). Some bacteria in CAMAs may be facultative intracellular organisms. For example, gram-negative bacilli, such as Endozoicomonas recovered from CAMAs, possess relatively large genomes ranging from 5.6 to 6.9 million base pairs (Mbp) (Maire et al., 2023) indicating it to be a facultative symbiont, because obligate endosymbionts typically have much smaller genomes (<1.5 Mbp) (Darby et al., 2007). The evidence of Endozoicomonas exhibiting diverse aggregation patterns, ranging from contained aggregates to irregular shapes lacking clear boundaries, also supports the hypothesis that some bacteria within CAMAs may be facultative intracellular organisms (Gotze et al., 2024). The difficulty in culturing bacteria from CAMAs and their cnidarian host cells presents a substantial challenge in understanding their role in coral health, their colonization processes, and regulatory mechanisms. Historically, large advances in our understanding of host-microbe interactions have stemmed from the ability to manipulate both bacteria and their hosts. For example, studies on Wolbachia in insects have illuminated how this symbiont influences reproduction and population dynamics (Fallon, 2021), whereas research on the symbiosis between squid and Aliivibrio fischeri has provided insights into bacterial colonization and bioluminescence (Nyholm & McFall-Ngai, 2021). Future research could focus on developing new methods for culturing CAMAs, such as culturomics (Vanstokstraeten et al., 2022) and in vitro cultivation of bacteria from CAMAs with coral primary cells (Nowotny, Connelly & Traylor-Knowles, 2021).

The presence of CAMAs in apparently normal corals reported by others (Peters, Oprandy & Yevich, 1983; Wada et al., 2019; Wada et al., 2022) and the presence of CAMAs in visually normal fragments in our study indicate they may play an important role in coral physiology, akin to mutualistic intracellular bacteria in insects that influence nutrition, immunity, and evolution (Eleftherianos et al., 2013; Coolen, Magda & Welte, 2022). Bacterial endosymbionts in insects are vital for maintaining host health, particularly in the face of emerging diseases, environmental stress, and climate change (Vásquez et al., 2023). For example, Buchnera bacteria provide essential amino acids to aphids, allowing them to subsist on nutrient-poor diets like plant sap (Gündüz & Douglas, 2009). Given these parallels, it seems reasonable to hypothesize that CAMAs may play analogous roles in corals. The genomic functional characterization of Endozoicomonas in marine hosts indicates Endozoicomonas within CAMAs may have a wide-range symbiotic spectrum from mutualism and commensalism to opportunism and parasitism (Pogoreutz & Ziegler, 2024). Endozoicomonas exhibit aggregative behavior in the gill and digestive epithelium and have been associated with parasitic and pathogenic relationships with fish and clams (Katharios et al., 2015; Bennion et al., 2021). In contrast, we observed no associated host cell pathology. The squid-Vibrio model (McFall-Ngai, 1999) might serve as an informative procedural analogue towards better understanding how CAMAs interact and beneficially or adversely affect the coral host.

Conclusions

This study of CAMAs in diseased corals using morphology and Gram staining revealed morphological differences in the bacteria found in individual CAMAs, ranging from coccoid to bacillus-shaped and from gram-negative to gram-positive, highlighting complexity of CAMAs. Corals affected by algal overgrowth had fewer CAMAs compared to those with other types of lesions, which indicates that CAMAs may be involved in the microbial community shifts associated with nutrient pollution and increased algae cover. Further, geographic variations in CAMA abundance were found in corals from the Island of Hawai‘ i having significantly higher numbers compared to O‘ ahu, potentially reflecting anthropogenic effects that are much greater on the densely populated island of O‘ ahu. Future research could focus on identifying the specific microbial species within CAMAs. Confirming CAMAs’ abundance and complexity between healthy and diseased coral, especially algae overgrowth, could highlight the potential role and dynamics of CAMAs in coral health.

Supplemental Information

Supplemental Information 1 Raw data of each individual cell-associated microbial aggregate (CAMA)

Each data point includes case number, animal, location, date of collection, gross lesion type, anatomical and tissue location of cell-associated microbial aggregate (CAMA), Gram-stain characteristics, shape of bacteria, tissue area measured (mm2), number of CAMA counted, and CAMA/cm2.

Supplemental Information 2 Raw data of enumeration of paired normal/lesion fragments from given coral colonies

The data partitioned by case number, animal, location of collection, tissue area assessed, cell-associated microbial aggregates (CAMA) count, CAMA/cm2, coral colony ID and lesion status. Note some fragment listed in File S1 were excluded from File S2 because a complete tissue pair (normal and lesion) was not available.

The use of trade, firm, or product names is for descriptive purposes only and does not imply endorsement by the US Government. We would like to thank Robert Rameyer, Renee Breeden, and Julie Tilley for helping with coral collecting and preparing tissue for histology slides as well as members of the ToBo lab for suggestions on this work. ChatGPT (OpenAI) was used to assist with debugging R code for data processing and plotting but not for data generation or analyses.

Additional Information and Declarations

Competing Interests

Author Contributions

Field Study Permissions

Data Availability

Robert J. Toonen is an Academic Editor for PeerJ.

Chutimon Singhakarn conceived and designed the experiments, performed the experiments, analyzed the data, prepared figures and/or tables, authored or reviewed drafts of the article, and approved the final draft.

Robert J. Toonen conceived and designed the experiments, authored or reviewed drafts of the article, and approved the final draft.

Thierry M. Work conceived and designed the experiments, authored or reviewed drafts of the article, and approved the final draft.

The following information was supplied relating to field study approvals (i.e., approving body and any reference numbers):

Corals samples were collected under Hawaii Department of Aquatic Resources Permit (SAP2025-28).

The following information was supplied regarding data availability:

Raw data is available in the Supplementary Files and at USGS:

Singhakarn, C., Toonen, R.J., and Work, T.M., 2025, Gram staining of cell-associated microbial aggregates in coral of the Pacific Ocean from 2001 to 2021: U.S. Geological Survey data release, https://doi.org/10.5066/P1S67TIJ.

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
