# Peer review of "Gram staining reveals diverse bacterial associations in coral cell-associated microbial aggregates in the Pacific Ocean"

_PeerJ, doi:10.7717/peerj.19867_

## Round 0.1 · original submission · Major Revisions

Summary of reviewer comments (see below for detailed comments):
- Inclusion of more biological discussion in the abstract;
- Inclusion of why different bacterial species might aggregrate into CAMAs (references to potential interactions, including cross-feeding) in the discussion,
- Gentle rephrasing of the title to not overemphasize the (unsupported) presence of Endozoicomonas;
- Inclusion of the spatial aspect of gram staining to strengthen the study justification;
- Suggested inclusion of relevant literature for a more balanced discussion;
- Validity of the statistical approach (Shannon Index vs. Chi Squared) and (incomplete supplementary data).
- Please note that one reviewer has raised concerns regarding the assessment of the gram staining analysis. It will be helpful to either provide a validation of the gram staining approach (please refer to the detailed comments by reviewer 3), or using a more objective approach for gram stain evaluation, for instance via image analysis.

In addition, some minor comments from my side:

- Not all Endozoicomonas are motile in culture (for instance, please refer to the original description of E. atrinae: https://doi.org/10.1099/ijs.0.060780-0) – hence the statement in lines 225 should be slightly revised.
- The genomes of Endozoicomonas have a greater range than the indicated 5.9 – 6.9 Mbp – the smallest being Ca. Endozoicomonas endoleachii with 2.5 Mbp (Hyams et al. 2023 Frontiers in Microbiology; https://doi.org/10.3389/fmicb.2023.1072053) and the largest being Endozoicomonas marisrubri 6c with 7.9 Mbp (at 97 % completeness; Pogoreutz et al. 2022 ISMEJ; https://doi.org/10.1038/s41396-022-01226-7). I suggest gently revising this sentence if the statement on genome sizes is retained in the manuscript.

Reviewer 1 ·

Basic reporting

General comments

Singhakarn, Toonen and Work provide a descriptive study of CAMAs abundance and colonizers (according to gram staining and cell shapes) in three key reef building corals. The manuscript is written in clear and easy-to-understand English, and the structure, figures, and supplementary material are appropriate to the study. I suggest italicising the coral species name in the figures. The experimental design and methodologies fit the purpose of the study well. The abstract lacks flow, especially the last few sentences. I suggest the authors tie it together with more biological insight instead of only listing their results. Methods and Results are well laid out. In the discussion section, I expected to see some text (even speculative) on the possible reasons why different bacterial species aggregate into CAMAs.

Detailed comments

90: reference is missing

94-95: is it? Why? What do CAMAs do for the coral holobiont health?

123-126: can the authors describe how the coral species were identified?

206: do the authors mean “four types of bacteria morphologies were observed in CAMAs”?

238: reference missing.

Experimental design

no comment

Validity of the findings

no comment

·

Basic reporting

Lines 1-3: The study attributes many Gram-negative rods in CAMAs to Endozoicomonas based on morphology and prior literature. However, Gram staining alone is insufficient to confirm bacterial identity. Without molecular analyses (e.g., 16S rRNA sequencing or FISH), the possibility of other rod-shaped bacteria previously detected in CAMAs (e.g., Aquarickettsia in pharyngeal tissue DOI: 10.1111/1462-2920.15245 and Simkania in tentacles DOI: 10.1126/sciadv.adg0773) suggests a more complex microbial community.
To reflect uncertainty in bacterial identification, revise the title to avoid implying Endozoicomonas was definitively identified in all CAMAs, e.g., :
- "Gram staining suggests greater complexity beyond Endozoicomonas in coral cell-associated microbial aggregates."
- "Gram staining reveals diverse bacterial associations in coral cell-associated microbial aggregates."

The manuscript is scientifically sound and written in professional English but would benefit from a final round of proofreading to address grammatical errors and improve readability. Key corrections include:
Line 3: ‘Ocean’ should be capitalized as part of the proper noun Pacific Ocean.
Line 50: Add a comma after "nor" in "paired normal and lesioned tissues, nor among lesion types" to improve readability.
Line 58: Improve the last sentence for clarity: "other bacterial species, in addition to Endozoicomonas, are colonizing CAMAs.
Line 59: ‘Ocean’ should be capitalized as part of the proper noun Pacific Ocean.
Line 66: Correct “bacteria-coral interaction" to "bacteria-coral interactions" as there are multiple types of interactions occurring between bacteria and corals.
Line 71: "Porites manifesting tissue loss" → "Porites experiencing tissue loss”. "Manifesting" is unnatural in this context; "experiencing" is clearer and more appropriate.
Line 80: "in situ" should be italicized.
Line 81: "other microbes are involved in CAMAs formation" → "other microbes are involved in CAMA formation”. The correct possessive structure.
Line 82: each CAMAs → each CAMA. “Each" refers to a singular noun, so it should be "each CAMA" (not "each CAMAs").
Line 89: "within single CAMA" → "within a single CAMA”. “Within single CAMA" is missing the article "a" for grammatical correctness.
Line 90: "CAMA sample" → "CAMA samples”. “Sample" should be plural because multiple CAMAs were analyzed.

Line 92: "CAMAs complexity" → "CAMA complexity”. As “CAMAs" is plural, but "complexity" is singular, the correct phrasing is "CAMA complexity" (or "the complexity of CAMAs").

Line 97: "provides presence or absence of DNA" → "indicates the presence or absence of DNA”. “Perhaps “indicates" is a better verb choice.
Line 104: Down-stream" should be one word: "downstream".
Line 105: “Lead to use" → "lead to the use" (missing article) & “bacteria-type" → "bacterial type" (grammatically correct form).
Line 115: "Remove unnecessary comma in 'Acropora, has' → should be 'Acropora has'."
Line 128: "Corals samples" → "Coral samples”. Typo
Line 133: "trimmed and processed for standard histology method as described" → "trimmed and processed following standard histological methods as described (Work & Aeby, 2010).
Line 143: "Gram stained slides" → "Gram-stained slides”. “Gram-stained" should be hyphenated since it modifies "slides".
Line 150: "Bacteria shape within CAMAs" → "Bacterial shape within CAMAs" (plural)

Line 225: "Most CAMAs were filled with Gram-negative rods, morphologically consistent with the genus Endozoicomonas, which in culture presents as Gram-negative motile rods and is abundant across marine invertebrates and fish." Added comma for correct clause separation.
Line 230: "Gram-positive bacteria in coral tissues are extensively documented using both culture-dependent (Sweet et al., 2021) and culture-independent (Ainsworth et al., 2015) methods. "Have been" is more appropriate than "are" for discussing past studies.
Line 238: "Bacteria" → "bacterium" (singular).
Line 262: CAMAs samples" → "CAMA samples" (incorrect pluralization). Alternatively, remove the word “samples”
Line 286: Consider rephrasing sentence for better readability, e.g., "Some bacteria in CAMAs may be facultative intracellular organisms. For example, Gram-negative rods, such as Endozoicomonas recovered from CAMAs, possess relatively large genomes ranging from approximately 5.9 to 6.9 million base pairs (Mbp) (Maire et al., 2023)."

Line 314: Consider rephrasing sentence for better readability, e.g., "While the relationship between nutrient levels, algal overgrowth, and CAMA abundance remains unclear..."
Line 319: "CAMAs presence" → "The presence of CAMAs" (correct noun phrase structure).
Line 356: Consider rephrasing sentence for better readability, e.g., “A study of CAMAs in diseased corals using morphology and Gram staining revealed morphological differences among individual CAMAs, ranging from coccoid to rod-shaped and from Gram-negative to Gram-positive, highlighting the complexity of CAMAs."

Line 363: “CAMAs abundance" → "CAMA abundance" (correct singular/plural agreement).
Line 364”: algae overgrowth could highlight potential role and dynamic of CAMAs → algal overgrowth could highlight the potential role and dynamics of CAMAs.

Experimental design

Line 147 & Figure 6: The skeleton is categorized as a tissue layer alongside the epidermis, mesoglea, gastrodermis, and calicodermis. However, the skeleton is a non-living structural component secreted by the calicodermis, not a tissue layer. I recommend revising this classification to reflect coral anatomy accurately. For example:"Distribution of CAMAs was categorized by tissue layers (epidermis, mesoglea, gastrodermis, and calicodermis) and by skeletal localization (within or adjacent to the skeleton)."
This clarification would improve anatomical accuracy and align with established coral biology terminology.

Lines 161–170: The manuscript does not specify the version numbers of RStudio and its associated packages. Including version details for ggstatplot, vegan, and others would ensure reproducibility.

Figure 3: The term "growth anomaly" is unclear to the reader; it appears as polyps in the shaded part of the colony. Providing a clearer definition or visual example would improve understanding.

Figure 4: It would be helpful to include the sample size (n=x) for each geographic location in the figure or caption. This would clarify how many coral samples were studied from each region, as counting from the violin plot is not intuitive.

Figure 5: The term "external observation" requires clarification. Does this refer to the health state of the whole coral colony, or were observations based on histological sections of coral fragments?

Validity of the findings

Line 226: The reference to (Schreiber et al., 2016) focuses on Endozoicomonas in ascidians. To better support the claim, additional references, such as DOI: 10.1016/j.tim.2023.11.006, could be cited.

Line 245: While Gram staining is appropriately highlighted as a diagnostic tool, consider emphasizing its additional advantage: spatial localization of bacterial members within tissues. Unlike 16S rRNA sequencing, Gram staining provides anatomical context, which strengthens the justification for its use in this study.

Lines 218–219: The observation of coexistence of bacteria with differing morphologies and Gram staining characteristics within the same CAMAs is novel and significant. This finding suggests greater microbial diversity within aggregates than previously recognized. I recommend emphasizing this discovery in the discussion.

Lines 280–290: The discussion raises an interesting hypothesis about the facultative intracellular nature of Endozoicomonas, supported by its relatively large genome size. To strengthen this claim, consider referencing the study "Differential aggregation patterns of Endozoicomonas within tissues of the coral Acropora loripes" (DOI: 10.1101/2024.12.17.629048). This study aligns with the hypothesis by showing that Endozoicomonas can exhibit diverse aggregation patterns within coral tissues.

Lines 297–299: When suggesting alternative methods for studying obligate or unculturable symbionts, coral cell culture systems should also be mentioned. The study "Primary coral cell cultures as a tool to study coral–microbe interactions" (DOI: 10.1038/s41598-021-83549-7) provides a useful framework for maintaining such symbionts in vitro. Including this reference would broaden the discussion of potential research avenues.

Lines 318–321: The manuscript suggests that the presence of CAMAs in visually normal fragments supports their potential role in coral physiology. However, as stated in line 305, all fragments analyzed in this study were from diseased colonies. This raises the possibility that CAMAs may function differently in diseased versus healthy corals. Discussing the spectrum of Endozoicomonas interactions—from mutualism to parasitism—(DOI: 10.1016/j.tim.2023.11.006) could add depth to the interpretation. Are CAMAs always indicative of health, or could they also reflect disease states?

Additional comments

This manuscript provides a comprehensive and well-structured study on the composition, distribution, and potential role of cell-associated microbial aggregates (CAMAs) in coral health. The study is scientifically sound and addresses an important topic within coral microbial ecology, offering valuable insights into microbial associations within coral tissues. The authors employ robust sampling and histological methods, contributing significantly to our understanding of CAMAs across different coral species and geographic regions.

The paper is clearly written and well-referenced, providing a strong foundation for the study. The findings are relevant to the field and underscore the potential complexity and diversity of CAMAs.
Overall, this is a valuable contribution to the field of coral microbial ecology, and with the suggested revisions, it will provide an even stronger foundation for future studies on the ecological and functional roles of CAMAs in coral health and resilience.

Reviewer 3 ·

Basic reporting

-Clear and unambiguous, professional English used throughout. - Yes

-Literature references, sufficient field background/context provided. - Yes

-Professional article structure, figures, tables. Raw data shared. - Yes, but raw data files are incomplete; see comment below.

-Self-contained with relevant results to hypotheses - Yes, though no specific hypothesis given.

Experimental design

-Original primary research within Aims and Scope of the journal. - Yes

-Research question well defined, relevant & meaningful. It is stated how research fills an identified knowledge gap. - Yes

-Rigorous investigation performed to a high technical & ethical standard. – No, need more detail regarding validation of methods. See specifics comments in Section 4 Additional Comments.

-Methods described with sufficient detail & information to replicate -No, need more detail regarding validation of methods. See specifics comments in Section 4 Additional Comments

Validity of the findings

-Impact and novelty not assessed. Meaningful replication encouraged where rationale & benefit to literature is clearly stated. - Yes

-All underlying data have been provided; they are robust, statistically sound, & controlled. – No. One of the raw data files are incomplete; see comment below. Also, only representative photos of Gram-stained CAMAs are provided. Given the subjective nature of assessing Gram-reaction, I recommend that the authors provide links to a repository of all photos so that the reader can also assess the Gram-reaction. See specifics comments in Section 4 Additional Comments

-Conclusions are well stated, linked to original research question & limited to supporting results. – Yes, but some conclusions may change if data is subject to a repeat of determination of Gram-reaction of each CAMA. See specifics comments in Section 4 Additional Comments

Additional comments

-Line 97: Just FYI: by comparing propidium monoazide (PMA) treated versus non-treated samples, metagenomics (or other molecular techniques) can distinguish between bacterial DNA signatures derived from viable vs. non-viable cells. I suggest including this technique here in the introduction and also scanning the literature to see if anyone has done this type of comparison on bacteria in CAMAs.

-Lines 132 - 137: I see in Lines 245 – 263 of the Discussion section that the authors do address some of the potential errors of Gram-staining, but I do not think these information is sufficient to address my concerns about the data. Has this protocol (decalcification and then Brown Hopps method for Gram-staining) been validated for coral-associated bacteria or used in previous studies? Did more than one person examine each slide and decide the Gram-reaction to examine Inter-observer Reliability? Did you try any color-metric analyses using photo-editing software to try to get a more objective measure of color and hue? Is it possible that the scanning process on the HP Color LaserJet may have influenced any visual assessment of color? I ask for more information and validation because my personal assessment of Gram-reaction in Panels A and B of Figure 8 do not agree with the stated assessment listed in the caption, see comment regarding Figure 8 below. It’s important to provide some additional data to validate this protocol and/or to bolster your assessment of each Gram-reaction. This additional data is imperative because a good chunk of your conclusions rely on these visual determinations.

-Line 169: Not sure if Shannon’s index is a particularly informative statistic to compare bacterial cell morphology, given that you are only categorizing into four groups (Gram-reaction x rod/cocci). Perhaps a chi-squared would be better to compare cell morphologies within CAMAs across species/health states/site/tissue/etc.

-Lines 232 – 243: Corals harbor a great diversity of Gram-positive bacteria that belong to the Actinobacteria but also the Firmicutes Phyla, so it’s unclear why the authors are pointing out Candidatus actinomarina and Spiroplasma specifically. The authors do not document any actinobacterial hyphal forms in their data set, and the authors have no evidence that Spiroplasma is present in their corals. I recommend that the authors remove this information or alter language to be broader and more inclusive about other potential gram-positive bacteria that could be within the examined CAMAs.

-Lines 245 – 263: I think much of this information should be moved to the methods section.

-Lines 261 – 263: It’s good that a Gram-positive control was included, but why not a Gram-negative control?

-Figure 2: Is there any way to make the legend, or at least the color dots in the legend, of Figure 2 bigger? It’s difficult to distinguish between the of the dots representing ‘Apparently Normal’ and ‘Bleaching’.

-Figure 8: I do not agree with the assessment of Gram reaction in some of the representative photos in Figure 8. In Panel A, many of those rods seem purple to me, which I would classify as Gram-positive, but the caption classifies as Gram-negative. In Panels B and C, to me, there seems to be a mix of purple and pink cocci, but the caption classifies as just Gram-negative.

-Supplementary File 2: It seems that there is missing data in Supplementary File 2. For example, Line 123 states 87 fragments were examined, but there are only 70 lines of data in Supplementary File 2 which represents data by coral fragment. Lines 124 (and Table 1) states that there were 4 colonies of Acropora cytherea examined, but Supplemental File 2 only shows 3 colonies of this species (colony IDs 7, 8, and 32). Numbers for colonies examined as listed in methods vs as listed in Supplemental File 2 also do not match up for the other coral species. If some coral fragments were not included in the examination (and thus why they are not listed in Supplemental File 2), need to clarify why.

---

## Round 0.2 · Minor Revisions

Thank you for your patience. Unfortunately, not all reviewers from the previous round of evaluation were available. The new reviewers are very much in favour of your manuscript, but have highlighted a few points from which your work could benefit.

These include (but please refer to more detailed feedback by the expert reviewers):

- Double-check discrepancies of coral colonies listed between table 1 and supplement.
- Please consider providing all gram-stained images in a repository accessible to reviewers and readers.
- Minor amendments (typos, inconsistencies) throughout the manuscript as suggested by the reviewers, especially as highlighted in the annotated manuscript as uploaded by reviewer 3.
- Clarify certain procedures, such as different sets of histology slides.
- Consider minor modifications in their figures, as suggested by the reviewer (e.g. add a jitter in Fig. 5).
- Add relevant literature to support certain statements (please refer to annotated manuscript by the reviewers).

Thank you for submitting your work to PeerJ.

·

Basic reporting

The revised manuscript provides valuable insights into the spatial and morphological diversity of cell-associated microbial aggregates within coral tissues. The authors have clearly improved the manuscript in response to reviewer comments from the first round. The methods remain robust, the data are compelling, and the revised text reads more clearly and cohesively. I believe the authors have addressed the majority of the concerns, and I support publication of the manuscript.

That said, I have a few minor points for consideration:

Thank you for clarifying the two datasets (lines 167–176). The revised explanation improves transparency, but I still suggest explicitly stating in the manuscript (or Supplementary File 2 caption) why some colonies listed in Table 1 (e.g., one Acropora cytherea colony) are missing from Supplementary File 2—i.e., whether they were excluded due to a lack of a lesion pair, missing data, or another reason. This would eliminate residual ambiguity and help readers understand how samples were selected for each analysis.

Additionally, the manuscript would benefit from one final round of proofreading to improve grammatical consistency and fluency. For example, there are occasional inconsistencies in singular versus plural usage of “CAMA” and “CAMAs,” which are sometimes used interchangeably, even when referring to multiple aggregates (e.g., lines 185 and 187). A careful read-through to correct such minor issues would enhance the clarity of the manuscript.

Experimental design

no comment

Validity of the findings

no comment

Additional comments

no comment

Reviewer 4 ·

Basic reporting

The language is clear throughout, however there are some typos which need to be addressed throughout the text (see comments below for examples).
The literature and context is sufficient, relevant and up to date.
The article structure is professional however the figures could be modified slightly to be easier to interpret and more professional. The supporting data appears to be present in it’s entirety.
The results and conclusions are supported by the data and the hypotheses are addressed.

Experimental design

This research is original and falls within the scope of the journal. They clearly state how this research fills a knowledge gap and explain the benefit of this methodology for the purpose of answering their question. There are some instances where the methods could be expanded in order to help the reader understand better what samples were used for what analysis (see specific comments below). The investigation was rigorous and performed to a high technical and ethical standard. The methods were explained well enough for replication.

Validity of the findings

The data are all present to the degree that the statistical analyses could be carried out again. I do believe that it would be nice to have a repository of all of the gram-stained images for validation. The authors have clearly considered the appropriateness of the statistical analyses based on the data and the assumptions of the models.
The conclusions are clearly described, drawn directly from the results and are not overstated.

Additional comments

General comments
This is a nicely laid-out, well explained and interesting manuscript. It is clear that the authors took on board the previous reviewers comments and greatly amended their manuscript. I believe there are still some minor amendments to be made to clear up all typos, improve the figures and to generally polish the manuscript. I believe that the work is important, and they do not overstate their findings, but give a balanced conclusion which is beneficial to the field.
Introduction -

Line 69 perhaps mention Cnidaria generally as you refer first to an anemone and then to corals with no explanation of the phylogenetic relationship between these taxa. This may be unclear to some readers.

Line 82-CAMA's'
Line 100-Anotomic'al'
Line 123-'reef building' coral taxa
Line 126-Coral identification 'was' based on ...

Methods - Is it relevant to the manuscript to describe the coral morphologies to this degree for the context of this paper?

Line 186-176. It would be useful to clearly describe the purpose of the two sets of histology slides and what questions they were used to answer.

Results -

Figure 2-

• the colours of point's'
• CAMA's'/cm² in individual... (CAMAs is often missing the s – check manuscript thoroughly for this) i.e. also in figure 3, 4 & 5.
• comma in the wrong place before "excluding outliner". The same typo is in Figure 3, 4 & 5 and instances of a space needing to be removed before a full stop or in brackets.)
• should be "outlie'r' not outliner,
• coral specie's'

Figure 4.
- I recommend that the authors move the boxes with the median values so that they do not obscure the data. I also believe that the statistical results can be greatly simplified in the figure and the full results presented only in the main text of the results (same comment for figure 5).
- For figure 4 and 5, it I suggest the authors either reduce the width of the box plots to fit inside of the violin plots, or preferably just have violin plots with a median and quartile lines within rather than a box plot overlayed. This would make it easier to interpret the data.

Figure 5

- I believe it was a good idea to jitter the data points in these figures and encourage the authors to do the same for figure 5.
Figure 6
- Remove the ‘In’ A / ‘In’ B
- This is perhaps a personal preference, but I suggest that the authors write the observations in the main text of the results rather than in the figure legend (same for figure 7).
Figure 7 –
- Title of figure legend is incomplete i.e. – Gram-stain images of confirmed using – there are words missing between confirmed and using.
- B – the numbers representing the bars on the y axis are confusing in this figure. I suggest they are either explained or changed to something more easy to understand.

Discussion
- I recommend adding a sentence to describe what the objective of the study was at the start of the discussion to remind the reader.
- Line 262- I think this sentence could be edited to be clearer i.e. 74% reduction in Porites with the condition bleaching with tissue loss versus healthy colonies..
- Line 325 – diversity ‘between’ and within
- Line 330 – can you really determine the diversity of a CAMA by gram staining? It could be that there are different bacterial species with the same morphology and gram – type.
Conclusion
- I recommend that the authors don’t start the conclusion sentence with A. Perhaps something like ‘This study’.

·

Basic reporting

The paper was prepared as appropriate for this journal and contained the hypotheses and relevant results. Although professional English was mostly used throughout, the authors had mistakes on spelling, capitalization, and punctuation, which are now indicated on the PDF of the manuscript uploaded. References and sufficient information on the background and context of the research were provided. However, this reviewer did note that a a few references should have been provided that better supported the claims in the text.

Experimental design

This study is appropriate for publication in PeerJ. The research question is well-stated and relevant, with supporting text explaining how the study fills an identified knowledge gap. The investigation included quality assurance and control procedures and the methods were described with detail that should allow replication; however, the fixative solutions for the coral samples and bird organ samples needs to be added, as this may affect final interpretation.

Validity of the findings

The manuscript encourages replication and additional research, the data appear to be robust, statistically sound, and controlled, and the Conclusions section links to the original question and supporting results.

Additional comments

None, but see marked-up manuscript.

---

## Round 0.3 · accepted · Accept

The remaining points raised by the reviewers have now been addressed, and your manuscript is now ready for publication.